# Longitudinal Follow-Up Using the Heel Enthesitis Magnetic Resonance Imaging Scoring System (HEMRIS) Shows Minimal Changes in Heel Enthesitis Assessed in Spondyloarthritis and Psoriasis Patients

**DOI:** 10.3390/jpm12111765

**Published:** 2022-10-25

**Authors:** Nienke J. Kleinrensink, Wouter Foppen, Iris ten Katen, Emmerik F. A. Leijten, Pim A. de Jong, Julia Spierings

**Affiliations:** 1Department of Rheumatology & Clinical Immunology, UMC Utrecht, Utrecht University, 3584 CX Utrecht, The Netherlands; 2Department of Radiology, UMC Utrecht, Utrecht University, 3584 CX Utrecht, The Netherlands; 3Department of Rheumatology, Sint Maartenskliniek, 6500 GM Nijmegen, The Netherlands

**Keywords:** enthesitis, inflammation, magnetic resonance imaging, psoriatic arthritis, psoriasis, spondyloarthritis

## Abstract

Enthesitis is a common clinical feature of spondyloarthritis (SpA). For reliable assessment of enthesitis the Heel Enthesitis Magnetic Resonance Imaging Scoring System (HEMRIS) was developed. The aims of this study were to evaluate changes in HEMRIS over time and to evaluate whether these changes correlated with changes in clinical parameters. This single-center observational study followed patients with SpA and psoriasis, regardless of presence of clinical heel enthesitis, for two years. Clinical evaluation and ankle MRIs were performed annually. Changes in HEMRIS were compared at one-year intervals using the Wilcoxon signed-rank test. The association between changes in the HEMRIS with changes in clinical parameters was evaluated using Spearman’s correlation coefficient. In total, 38 patients were included. An increase in the inflammatory and structural HEMRIS was identified in, respectively, 12 (17.9%) and 4 (6.0%) patients in one-year intervals. We found non-significant changes in the HEMRIS during longitudinal follow-up. Changes in the HEMRIS did not correlate with changes in local or general disease activity. Our results show that MRI-findings of enthesitis assessed with HEMRIS changed in a small number of patients in a one-year interval in an observational setting. Changes in HEMRIS were not associated with changes in clinical disease activity.

## 1. Introduction

Enthesitis, defined as inflammation at the anatomical location where tendons, ligaments and joint capsules insert to bone, is a key clinical feature of spondylarthropathies, including psoriatic arthritis (PsA) and ankylosing spondylitis (AS) [1,2]. Recommended treatment strategies for enthesitis vary and may include non-steroidal anti-inflammatory drugs (NSAIDs), glucocorticoids and, in PsA, advanced treatment with disease-modifying anti-rheumatic drugs (DMARDs) [3,4]. However, a golden standard for evaluation of enthesitis is lacking. Clinical evaluation of enthesitis, which relies on the subjective assessment of pain after applying pressure at insertion sites, has limited sensitivity and specificity [5,6,7].

To monitor enthesitis at an early stage and more accurately, imaging techniques, such as ultrasound and MRI could be employed [8]. An advantage of MRI over ultrasound is that it allows for visualization beyond the bone cortex and can detect peri-entheseal osteitis or ‘bone marrow edema’ [1]. For assessment of enthesitis with MRI in PsA and AS patients with a focus on the heel region, the Outcome Measures in Rheumatology (OMERACT) group has developed and validated the Heel Enthesitis Scoring System (HEMRIS) [9,10]. Using the HEMRIS, inflammatory and structural features of heel enthesitis can be assessed at the area of the Achilles tendon and plantar fascia. However, the sensitivity of HEMRIS for change over time needs to be further established in longitudinal studies, both for inflammatory and structural HEMRIS pathologies [9]. So far, one study did not show any significant changes after treatment with secukinumab or placebo-secukinumab after 52 weeks [11].

Besides its potential for evaluation of treatment effects, HEMRIS could provide insights in the pathogenesis of PsA. It has been hypothesized that SpA originates at the enthesis [5]. Imaging of the enthesis could possibly identify individuals with subclinical enthesitis progressing from psoriasis to psoriatic arthritis at an early stage, or predict disease activity.

The aims of this study are to evaluate the sensitivity of HEMRIS for change during longitudinal follow-up and to assess whether e changes in HEMRIS are associated with changesin clinical disease activity in psoriasis (Pso), PsA and AS patients. As a secondary outcome, we compared HEMRIS in Pso patients with and without progression to PsA to assess if HEMRIS was predictive of later development of PsA.

## 2. Materials and Methods

### 2.1. Study Design and Patients

In this observational longitudinal study, carried out in an academic hospital, we included patients aged 18–65 years who were diagnosed with cutaneous Pso (diagnosed by a dermatologist, with PsA excluded by a rheumatologist), PsA (fulfilling Classification Criteria for Psoriatic Arthritis (CASPAR) [12]) and AS (fulfilling Assessment of SpondyloArthritis international Society (ASAS) [13] classification criteria). Patients were selected regardless of presence of clinical ankle enthesitis at the Achilles tendon or plantar fascia. Exclusionary criteria included current use of conventional or synthetic disease-modifying antirheumatic drugs (DMARDs) at time of inclusion. The current study focuses on longitudinal follow-up with HEMRIS, baseline MRI-results, in comparison with clinical parameters and PET-CT were described in our previous paper [14].

### 2.2. Ankle-MRI Protocol and Scoring

Both ankles were evaluated separately on a 3-Tesla MRI scanner (3T TX, Philips Healthcare, Best, The Netherlands) using a head coil. The MRI protocol included the following sagittal sequences: T2-weighted, T2-weighted Spectral Attenuated Inversion Recovery, T1-weighted SPIR (Spectral Presaturation with Inversion Recovery) before contrast, and T1-weighted SPIR after contrast. Baseline ankle MRIs were assessed by two independent musculoskeletal radiologists for inflammatory and structural features of heel enthesitis using the HEMRIS. The HEMRIS is a scoring system containing the following inflammatory and structural pathologies: intratendon hypersignal, peritendon hypersignal, bone marrow edema, bursitis (Achilles tendon only), tendon/aponeurosis thickening, enthesophyte, bone erosion and intra-tendon hypersignal on T1-Weighted images. All pathologies are scored on a scale of 0–3 (none/mild/moderate/severe). Examples of all grades of different HEMRIS pathologies are provided in an imaging reference atlas [10].

The maximum HEMRIS of 2 ankles is 24 for entheseal inflammation at the Achilles tendon, structural damage at the Achilles tendon and structural damage at the plantar fascia; 18 for entheseal inflammation at the plantar fascia; 42 for total entheseal inflammation; and 48 for total structural damage.

All HEMRIS subscores were averaged as suggested in the original HEMRIS publication [9]. Average HEMRIS subscores of 0.5, 1.5 or 2.5 were evaluated in a consensus meeting. Follow-up MRI-scans were evaluated for changes by a musculoskeletal radiologist (IK). Ankle MRI-results were compared at one-year-intervals: MRIs obtained at year 1 were compared to baseline, and MRIs obtained at year 2 were compared to year 1. Change in HEMRIS was calculated by subtraction. For comparison of HEMRIS with disease activity, HEMRIS of both ankles were summated.

### 2.3. Clinical Assessments

Psoriasis activity was measured using the Psoriasis Area and Severity Index(PASI) [15]. PASI > 10 was categorized as moderate to severe disease [16]. Disease activity in PsA was assessed with the validated composite outcome measure ‘minimal disease activity’ (MDA) [17]. MDA is achieved when 5 out of 7 of the following criteria are met: tender joint count ≤ 1, swollen joint count ≤ 1, PASI ≤ 1, patient pain visual analogue score (VAS) ≤ 15, patient global disease activity VAS ≤ 20, health assessment questionnaire ≤ 0.5 and tender entheseal points (measured using the Leeds Enthesitis Index [18]) ≤ 1. Disease activity in AS was assessed with the Bath Ankylosing Spondylitis Disease Activity Index (BASDAI) [19]. The threshold value for active disease was set at ≥4. Enthesitis at the site of the ankle was assessed by palpation for tenderness of the insertions of the Achilles tendon and plantar fascia into the calcaneus. MDA-status, categorized PASI, categorized BASDAI and clinical assessment of enthesitis were compared at one-year intervals on three levels (stable, increase of disease activity and decrease of disease activity).

### 2.4. Data Analysis

Non-normally distributed continuous data were reported using medians and interquartile ranges (IQRs). Categorical variables were reported in frequencies and percentages. The initial HEMRIS of Pso patients with and without clinical progression to PsA were compared with the Mann–Whitney U Test.

Patients were followed during two years. We performed pairwise comparisons of the HEMRIS and clinical outcomes in one-year intervals. Overall change in HEMRIS in one-year intervals was assessed using the Wilcoxon signed-rank test. The potential association of changes in HEMRIS with changes clinical disease activity was assessed using Spearman’s correlation coefficient. Differences in HEMRIS at baseline in Pso patients with and without progression to PsA were compared using the Mann-Whitney U Test. Due to the small sample size, we did not perform corrections for multiple measurements.

Patients that were lost to follow-up before the first follow-up visit were excluded from all analyses. Patients that were lost to follow-up after year 1 were only excluded from pairwise analyses comparing year 1 and 2. The predetermined significance level was set at *p* < 0.05. Statistical analysis was performed using SPSS version 26 (IBM SPSS Statistics, IBM Corporation, Armonk, NY, USA).

## 3. Results

### 3.1. Patients’ Characteristics

At baseline, 38 patients (76 ankles) were included. Baseline patient characteristics and MRI-results are described in detail in our previous paper [14]. For the current study, disease activity and MRI-results at inclusion and at year one are used as baseline values. Briefly, Pso patients had low disease activity and moderate-severe PASI scores were observed in only 5 (20.8%) patients at the baseline. PsA patients had ‘minimal disease activity’ at baseline in 45.8% (n = 11) (Table 1). At baseline, 22 (87.5%) AS patients had a BASDAI ≥ 4 (Table 1). Achilles tendon and plantar fascia enthesitis were observed in, respectively, eight (5.9.%) and six (4.5%) ankles assessed at baseline and at 52 weeks (pooled data; Table 1). At baseline, five patients (6.6%) were treated with a DMARD (Table 1).

Serial MRI observations were available for 36 participants at 2 time points (baseline and 52 weeks) and for 34 patients at 3 time points (baseline, 52 weeks and 104 weeks). At baseline, one ankle-MRI was excluded from analysis because of insufficient image quality due to failure of fat suppression. At 104 weeks, the Achilles tendon could not be examined on one MRI-scan due to an artifact. In total, 137 serial MRI observations of heel enthesitis were available for change in HEMRIS; 137 at the location of the Achilles tendon and 138 at the location of the plantar fascia. At patient level (summated left and right HEMRIS) the total number of serial MRI observations for change in HEMRIS is 68 at the location of the Achilles tendon and 67 at the location of the plantar fascia.

### 3.2. Change in Disease Activity during Follow-Up

In most patients, the overall mild psoriasis severity remained stable throughout the study. When compared at one-year intervals, a decrease in the PASI (disease activity no longer classified as ‘moderate-severe’) occurred in three patients (13.0%). No increases in PASI from mild to moderate-high disease activity were observed. Two (18.2%) patients with Pso developed psoriatic arthritis during the two years follow-up. Two Pso patients were lost to follow-up (Appendix A).

Activity of PsA increased in n = 3 (13.0%), decreased in n = 6 (26.0%) and remained stable in n = 9 (39.1%) patients in one-year observation intervals. One PsA patient was lost to follow-up, another was excluded from the analysis at the last time point since no MRI-scan was obtained (Appendix A).

Activity of ankylosing spondylitis, increased in 2 (8.3%) patients, decreased in 2 patients (8.3%) and remained stable in 19 patients (79.2%) during one-year observation intervals.

During follow-up, newly diagnosed clinical enthesitis was observed in four (3.2%) Achilles tendon entheses and six (3.2%) plantar fascia. Treatment with a DMARD was initiated in 6 (8.6%) patients during longitudinal follow-up (Appendix A).

### 3.3. Change in the HEMRIS during Follow-Up

Total HEMRIS for both ankles were low at the start of one-year intervals and after follow-up (median score ≤ 1, range 0–24 or 0–18; Appendix A). Increase in the inflammatory and structural HEMRIS (cut-off value: ≥1) were identified in, respectively, 12 (17.9%) and 4 (6.0%) patients during one-year follow-up intervals. No significant changes were observed in continuous HEMRIS during longitudinal follow-up (Figure 1). The HEMRIS subitems that changed most frequently during longitudinal follow-up were ‘peritendon hypersignal’ (in 13 plantar fascia and 16 Achilles tendons: Figure 2; Appendix A) and ‘retrocalcaneal bursitis’ (in 20 Achilles tendons: Figure 3; Appendix A). No new bone erosions were observed during longitudinal follow-up (Appendix A).

Changes in clinical disease activity, assessed with measures of general disease activity (Figure 4) and with measures for local disease activity at the enthesis (Figure 5) were not associated with changes in HEMRIS. Changes in clinical disease activity, in subgroups based on clinical diagnosis of Pso, PsA or AS, were not associated with changes in HEMRIS (Figure 6). No difference was observed in HEMRIS results at time of inclusion of the 2 Pso patients who developed PsA during longitudinal follow-up, in comparison to the 9 Pso patients that did not (Appendix A).

## 4. Discussion

In this prospective observational study in Pso and SpA (PsA and AS) patients with limited clinical disease activity at the enthesis, we evaluated whether ankle enthesitis on MRI, assessed using the HEMRIS, was sensitive to change during longitudinal follow-up. No significant differences were observed in the HEMRIS after a one-year follow-up intervals (Figure 1). The secondary objective was to evaluate whether changes in clinical disease activity, assessed with general measures of disease activity, or with clinical examination of enthesitis, were associated with changes in the HEMRIS. HEMRIS results were not associated with changes in general or local measures of disease activity (Figure 2, Figure 4, Figure 5 and Figure 6). Overall, results of this study indicate that the HEMRIS remains stable during longitudinal follow-up in an observational setting.

In the original publication of HEMRIS by the OMERACT group, change in inflammatory pathologies before and after anti-TNF therapy was evaluated in a group of SpA patients [9]. The standardized response mean of HEMRIS was 0.7, which is considered moderate. Since a clinical diagnosis of enthesitis was not mandatory for inclusion, the authors suggested that the responsiveness of HEMRIS would be ‘good in trials with baseline enthesitis as an inclusion criterion’. Subsequently, clinical heel enthesitis was an inclusion criterion for patients included in the ACHILLES trial. However, a post-hoc analysis of the ACHILLES trial showed little changes in HEMRIS in both patients treated with secukinumab or placebo, while a higher but non-significant proportion of patients treated with secukinumab had resolution of clinical enthesitis [11]. Our results with minimal changes in HEMRIS are in line with findings of the ACHILLES trial.

A possible explanation for the minimal changes in the HEMRIS in our study, is the low initial HEMRIS results (Appendix A), allowing for little room for improvement during longitudinal follow-up in an observational setting. The low HEMRIS results could possibly be caused by the selection of study patients: we included both Pso (skin disease only) and SpA patients, and clinical enthesitis was not an inclusion criterion. Clinical enthesitis was present in only 6% of ankles at the Achilles tendon and 5% of plantar fascia at baseline. At follow-up, clinical assessment of enthesitis remained stable in the vast majority of patients (Figure 5). A benefit of inclusion of patients without a clinical diagnosis of enthesitis is that it allows for evaluation of the clinical importance of subclinical MRI-findings. As reported in our previous paper on baseline results of the current study, subclinical MRI-abnormalities occurred in 62.9% of Achilles tendons and in 32.9% of plantar fascias [14]. Our results indicate that subclinical HEMRIS findings are of limited clinical relevance since the large number of subclinical MRI findings did not predict the development of clinical enthesitis or structural changes on MRI after one and two years.

The finding that minimal changes in HEMRIS are not associated with changes in general disease activity, could be considered unexpected, since enthesitis has been attributed a major role in SpA disease pathogenesis in the proposed ‘enthesitis model’ [2]. With the subdivision of general and local disease activity in the categories ‘stable disease’, ‘increase’ and ‘decrease’ in clinical disease activity, we potentially bypass the effects of continuous high disease activity in SpA on structural HEMRIS pathologies. However, the formation of new structural HEMRIS pathologies occurred in only 6.0% of patients in one-year intervals and did not include bone erosions (Appendix A). Another possible explanation for the lacking association of HEMRIS with general disease activity is that with HEMRIS, only the ankle is assessed for enthesitis. Imaging multiple entheses at once with MRI can be done by using whole body MRI, but this technique has its challenges, such as limited image resolution and highly varying interrater agreement for enthesitis lesions [20].

This analysis included patients with cutaneous Pso to assess if subclinical MRI changes at the enthesis could predict future progression to PsA. We did not find differences between HEMRIS at inclusion in Pso subjects with and without later development of PsA (Appendix A). The results of this explorative analysis must be interpreted with caution though, due to the low number of Pso patients included (n = 13) and progressing to PsA (n = 2). Further work is required to evaluate the potential of HEMRIS as an imaging biomarker for future onset of inflammatory joint disease in Pso.

Strengths of the current study include its longitudinal design, assessment of MRI-scans by experienced radiologists and the comparison of MRI-results with clinical assessments of both general disease activity and local disease activity at the enthesis. One limitation of the study is its relatively small sample size. Since this is an exploratory study, no sample size calculations were made, however this could increase the chance of a type II error. To increase power, we analyzed the data over one-year intervals. With a total follow-up duration of two years, each subject occurred in the dataset twice (regarding comparisons of the HEMRIS with general measures of disease activity), or four times (regarding comparisons of the HEMRIS with clinical examination of enthesitis at both ankles). Analyses were not adjusted for multiple measurements in patients because of limited power, however no significant changes in HEMRIS were observed in the unadjusted analyses. Another limitation is the observational study design. Further research in larger populations is required to further establish the responsiveness of HEMRIS to change and its association with change in clinical disease activity. In the ongoing TOFA-PREDICT study (EudraCT Number 2017-003900-28), we aim to investigate the effect of three different DMARDs (tofacitinib, methotrexate and etanercept) on HEMRIS results in a PsA population with more active disease.

## 5. Conclusions

MRI findings of heel enthesitis assessed with HEMRIS changed in a small number of patients in an observational setting. Although changes of enthesitis evaluated on MRI were minimal in this study, quantitative MRI assessment of enthesitis could potentially visualize change in these structures in great detail.

## Figures and Tables

**Figure 1 jpm-12-01765-f001:**
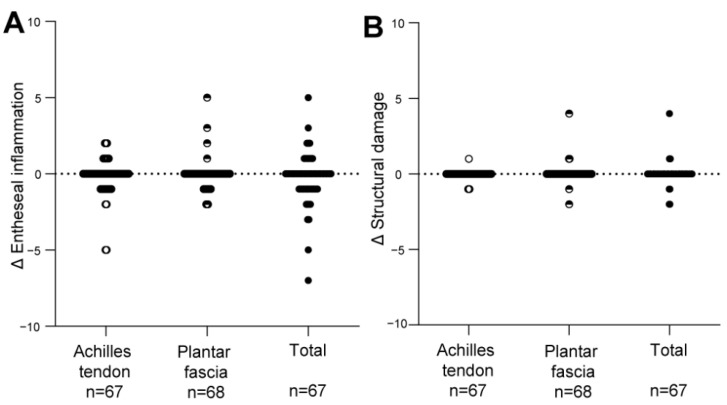
Change in HEMRIS Entheseal inflammation and HEMRIS Structural damage after longitudinal follow-up. After one-year follow up intervals, no significant changes were observed in (**A**) HEMRIS Entheseal inflammation, and (**B**) HEMRIS structural damage. Data are presented as the median change and individual values of change in HEMRIS (Delta HEMRIS). HEMRIS scores of both ankles were summated. Figure Legend: Abbreviations: Δ = delta, HEMRIS = Heel Enthesitis Scoring System.

**Figure 2 jpm-12-01765-f002:**
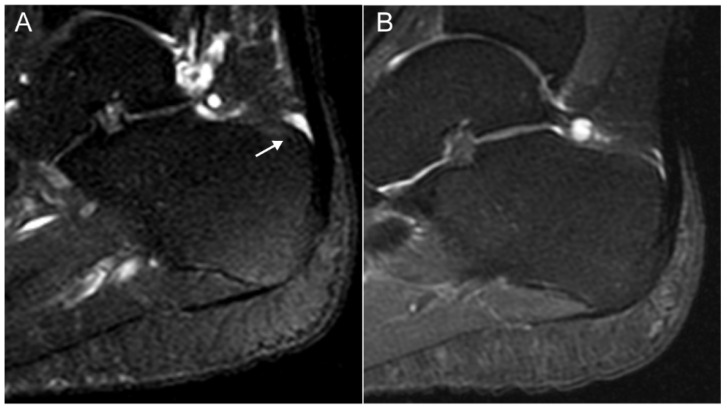
MRI images of a 48-year old female with ankylosing spondylitis, showing an increase in peri-Achilles tendon hypersignal after one-year longitudinal follow-up. (**A**) T2 SPAIR weighted image showing no hypersignal surrounding the Achilles tendon, (**B**) T2 SPAIR weighted image showing hypersignal (arrow: moderate; grade 2) surrounding the Achilles tendon, close to its insertion. Figure Legend: SPAIR = Spectral Selection Attenuated Inversion Recovery.

**Figure 3 jpm-12-01765-f003:**
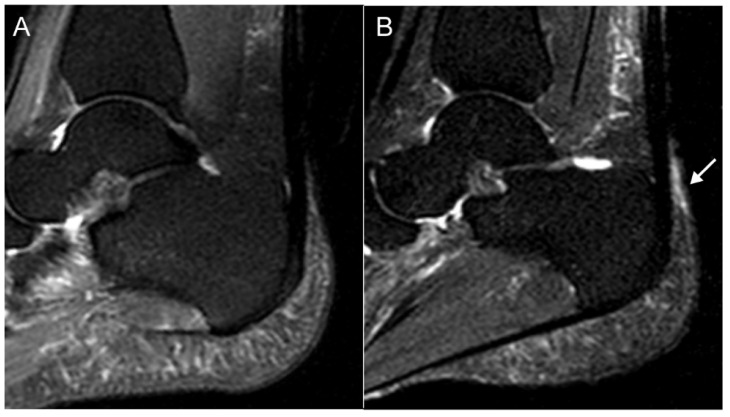
MRI images of a 33-year old male with psoriasis, showing a decrease in retrocalcaneal bursitis after one-year longitudinal follow-up. (**A**) T2 SPAIR weighted image showing hypersignal at the retrocalcaneal bursa consistent with mild inflammation at the bursa (arrow: grade 1; mild), (**B**) T2 SPAIR weighted image showing no hypersignal at the retrocalcaneal bursa. Figure Legend: SPAIR = Spectral Selection Attenuated Inversion Recovery.

**Figure 4 jpm-12-01765-f004:**
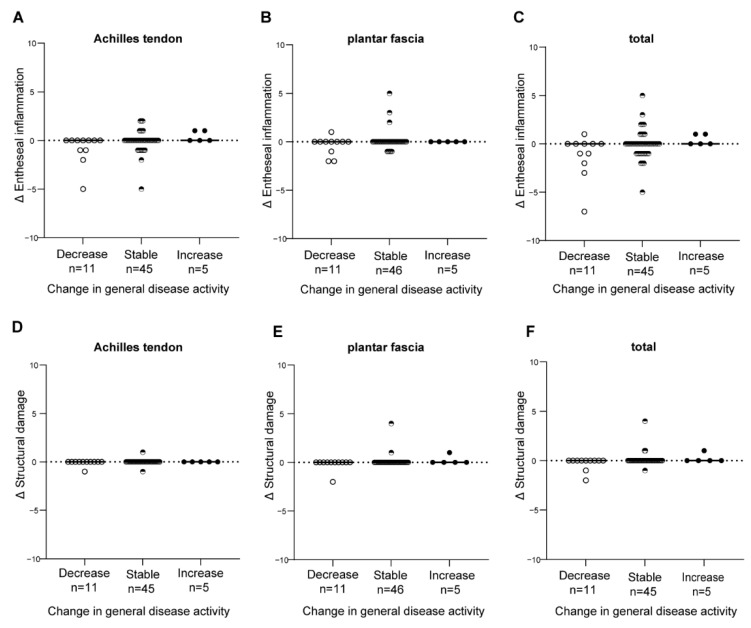
Changes in the HEMRIS in patients with different levels of change in general disease activity. Changes in general measures of clinical disease activity were not associated with changes in the HEMRIS Entheseal inflammation and structural damage at the area of the Achilles tendon (**A**,**D**), plantar fascia (**B**,**E**), and Achilles tendon and plantar fascia combined (**C**,**F**). Clinical disease activity results were pooled but were assessed differently in each patient category, using the PASI for psoriasis, MDA for psoriatic arthritis and the BASDAI for ankylosing spondylitis. Data are presented as the median change and individual values of change in HEMRIS (Delta HEMRIS). Figure Legend: Abbreviations: Δ = delta, BASDAI = Bath Ankylosing Spondylitis Disease Activity Index, HEMRIS = Heel Enthesitis Scoring System, MDA = minimal disease activity, PASI = Psoriasis Area and Severity Index.

**Figure 5 jpm-12-01765-f005:**
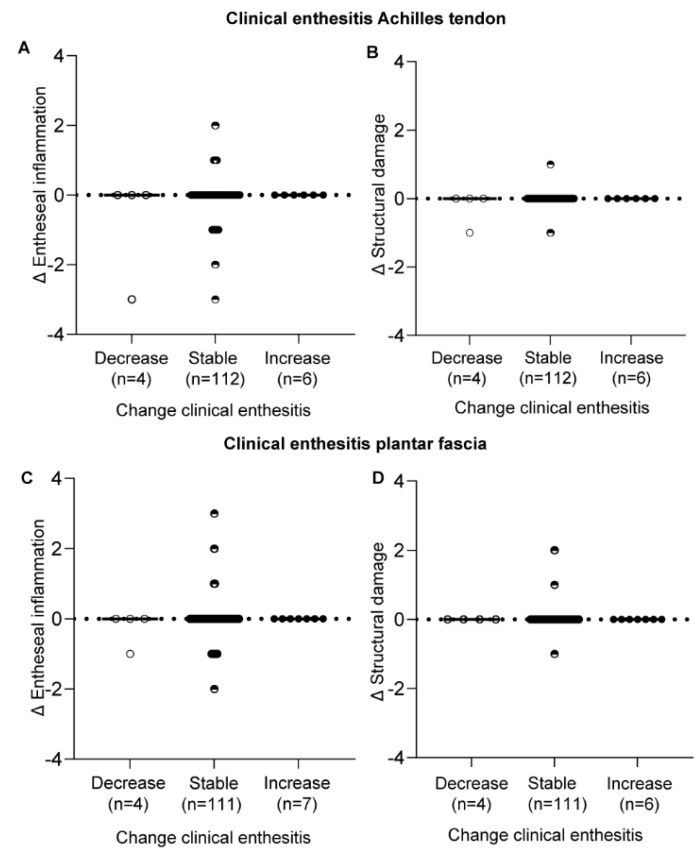
Change in the HEMRIS in patients with different levels of change in clinical enthesitis. Changes in clinical disease activity at the enthesis were not associated with changes in HEMRIS Entheseal inflammation at the Achilles tendon (**A**,**B**) and the plantar fascia (**C**,**D**). Data are presented as the median change and individual values of change in HEMRIS Entheseal inflammation and structural damage scores. Figure Legend: Abbreviations: Δ = delta, HEMRIS = Heel Enthesitis Scoring System.

**Figure 6 jpm-12-01765-f006:**
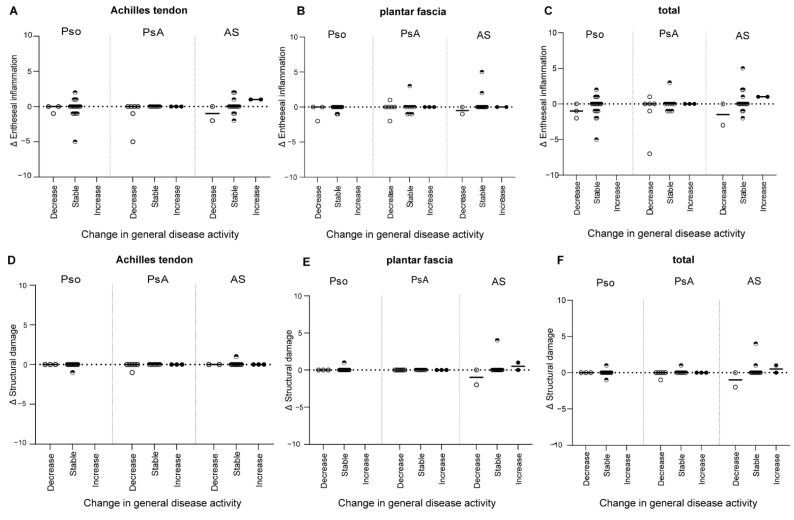
Changes in the HEMRIS in Pso, PsA and AS patients with different levels of change in general disease activity. Change in general disease activity was not associated with change in HEMRIS at the area of the Achilles tendon (**A**,**D**), plantar fascia (**B**,**E**), and Achilles tendon and plantar fascia combined (total HEMRIS; (**C**,**F**)), in patients with Pso, PsA and AS. Data are presented as the median change and individual values of change in HEMRIS (Delta HEMRIS). Figure Legend: Abbreviations: Δ = delta, HEMRIS = Heel Enthesitis Scoring System, Pso = psoriasis, PsA = psoriatic arthritis, AS = ankylosing spondylitis.

**Table 1 jpm-12-01765-t001:** Patients’ characteristics at baseline and 52 weeks (pooled data). Patients that were lost to follow-up were excluded from the analysis.

	Disease Category			
	Pso	PsA	AS	All
Total one year intervals, N	24	24	24	72
Demographics				
Male gender, n (%):	12 (50.0)	18 (75.0)	18 (75.0)	48 (66.7)
Age, median (IQR):	42.2 (34.7–53.4)	50.9 (40.6–52.9)	49.1 (38.8–52.2)	49.2 (36.8–52.8)
Disease duration in years, median (IQR)	22.5 (14.4–42.2)	7.8 (0.9–12.6)	8.7 (3.4–17.2)	NA
General disease activity:				
Pso: moderate-severe psoriasis, n (%)	5 (20.8)	NA	NA	NA
PsA: MDA, n (%)	NA	11 (45.8)	NA	NA
Missing, n (%)	NA	3 (12.5)	NA	NA
AS: BASDAI score ≥ 4, n (%)	NA	NA	22 (91.7)	NA
Medication:				
Current DMARD use, n (%):	0	4 (16.7)	1 (4.2)	5 (6.6)
Current NSAID use, n (%):	2 (8.3)	7(29.2)	16 (66.7)	25 (34.7)
Missing, n (%)	1 (4.2)	0	0	1 (1.4)
Inflammatory markers:				
ESR, median (IQR):	4.0 (2.0–10.0)	4.0 (2.0–6.0)	5.0 (3.0–6.0)	4.0 (2.0–6.5)
Missing, n (%)	2 (8.0)	0	1	2 (2.8)
CRP, median (IQR):	2.5 (0.9–5.7)	3.0 (1.6–4.6)	1.6 (0.9–4.4)	2.8 (1.2–4.5)
Missing, n (%)	2 (8.0)	0	1	3 (4.2)
Local disease activity at the enthesis:				
Achilles tendon, N enthesis	48	48	48	144
Clinical enthesitis, n (%)	2 (4.2)	1 (2.1)	3 (6.3)	6 (4.2)
Missing, n (%)	0	4 (8.3)	6 (12.5)	10 (6.9)
Plantar fascia, N enthesis	48	48	48	144
Clinical enthesitis, n (%)	1 (2.1)	4 (8.3)	2 (4.2)	7 (4.9)
Missing, n (%)	0	4 (8.3)	6 (12.5)	10 (6.9)

Table Legend. Abbreviations: AS = ankylosing spondylitis, BASDAI = Bath Ankylosing Spondylitis Disease Activity Index, CRP = C-reactive protein, DMARD = disease-modifying anti-rheumatic drugs, ESR = erythrocyte sedimentation rate, IQR = interquartile range, Pso = psoriasis, PsA = psoriatic arthritis, MDA = minimal disease activity, NA = not applicable, NSAID = non-steroidal anti-inflammatory drugs.

## Data Availability

All data relevant to the study are included in the article or uploaded as Appendix A. Data are available upon reasonable request. Please contact the corresponding author.

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
