# Peer review of "Longitudinal Follow-Up Using the Heel Enthesitis Magnetic Resonance Imaging Scoring System (HEMRIS) Shows Minimal Changes in Heel Enthesitis Assessed in Spondyloarthritis and Psoriasis Patients"

_jpm, 2022, doi:10.3390/jpm12111765_

Round 1
Reviewer 1 Report
Comments to the manuscript entitled: Longitudinal follow-up using the Heel Enthesitis MRI Scoring System (HEMRIS) shows minimal changes in heel enthesitis assessed in spondyloarthritis and psoriasis patients with ID: jpm-1955343. Congratulations to the authors of the manuscript. This is a great research about the changes of the heel enthesitis employing a test diagnostic as magnetic Resonance Imaging in patients with spondyloarthritis and psoriasis. The manuscript has a previous reference by the authors for a similar research in 2020 replicating some aspect of the previous research. Minor changes are required.
Title: please change MRI by Magnetic Resonance Imaging
Abstract: please add Magnetic Resonance Imaging by MRI
Keywords: delete “(MRI)”
1.- Introduction
Line 43.- please review what is (1).
Lines 45-49.- can authors explain better HEMRIS
2.- Material and methods
Can author explain how calculate the sample size?
Line 82.- Can authors add the HEMRIS subscores explanation despite the original reference?
Line 89.- Can authors explain what is PASI?
Results: Are properly describe
Tables: Please add abbreviations
Discussion
Can authors add more current references about the results of the research?
Please add a conclusion section apart from discussion and add limitations at the end of the discussion.
Author Response
Comments reviewer 1
Comments to the manuscript entitled: Longitudinal follow-up using the Heel Enthesitis MRI Scoring System (HEMRIS) shows minimal changes in heel enthesitis assessed in spondyloarthritis and psoriasis patients with ID: jpm-1955343. Congratulations to the authors of the manuscript. This is a great research about the changes of the heel enthesitis employing a test diagnostic as magnetic Resonance Imaging in patients with spondyloarthritis and psoriasis. The manuscript has a previous reference by the authors for a similar research in 2020 replicating some aspect of the previous research. Minor changes are required.
Title: please change MRI by Magnetic Resonance Imaging
Author response: as suggested by the reviewer, the title has been modified.
Abstract: please add Magnetic Resonance Imaging by MRI
Author response: as suggested by the reviewer, ‘MRI’ was replaced with the text ‘Magnetic Resonance Imaging’.
Keywords: delete “(MRI)”
Author response: as suggested by the reviewer, the abbreviation ‘MRI’ was deleted from the keywords.
1.- Introduction
Line 43.- please review what is (1).
Author response: there was an error in the formatting of the reference. We thank the reviewer for pointing this out and have adjusted this in the manuscript..
Lines 45-49.- can authors explain better HEMRIS
Author response: we thank the reviewer for this suggestion and explained HEMRIS in more detail with regard to clinical use and measures in the ‘Methods’ section.
2.- Material and methods
Can author explain how calculate the sample size?
Author response: since our study was an exploratory pilot study, no formal sample size calculations were made. Since this is a limitation, this is now mentioned in the ‘Discussion’ section.
Line 82.- Can authors add the HEMRIS subscores explanation despite the original reference?
Author response: the HEMRIS subscores have been added to the ‘Methods’ section.
Line 89.- Can authors explain what is PASI?
Author response: indeed, this abbreviation was not fully written out in our manuscript : PASI stands for the ‘Psoriasis Area and Severity Index’. This has now been adapted in the manuscript.
Results: Are properly describe
Tables: Please add abbreviations
Author response: thank you for pointing this out, we have now added a list of abbreviations to the Table.
Discussion
Can authors add more current references about the results of the research?
The research available on HEMRIS is limited and, to our knowledge, has been cited in our manuscript. We repeated a literature search on HEMRIS but did not find any new publications. If the reviewer has suggestions for any specific references we are happy to consider these.
Please add a conclusion section apart from discussion and add limitations at the end of the discussion.
Author response: we have made a separate conclusion section upon the suggestions of the reviewer. Also, the limitations are now listed in the last paragraph of the discussion.
Reviewer 2 Report
This is an interesting work. Some suggestions are enlisted below
1. Some important information about reference 12 should be reiterated instead of just mentioning " something has been described in our previous publication."
2. How do you determine the sample size of 38 patients (76 ankles) ? The information about sample size calculation should be described.
3. Please offer representative serial images with most apparent changes in "peritendon hypersignal" and "retrocalcaneal bursitis" from your series.
4. " Changes in clinical disease activity, assessed with measures of general disease activity (Figure 2) and with measures for local disease activity at the enthesis (Figure 3), were 168 not associated with changes in HEMRIS." " Changes in clinical disease activity, in subgroups 169 based on clinical diagnosis of Pso, PsA or AS, were not associated with changes in 170 HEMRIS (Figure 4)". "No difference was observed in HEMRIS results at time of inclusion of 171 the 2 Pso patients who developed PsA during longitudinal follow-up, in comparison to 172 the 9 Pso patients that did not." How do you make sure these findings are not underpowered ?
5. line 278 "visualizechange" should be corrected.
Author Response
Comments reviewer 2
This is an interesting work. Some suggestions are enlisted below
- Some important information about reference 12 should be reiterated instead of just mentioning " something has been described in our previous publication."
Author response: We thank the reviewer for this suggestion and we have described inclusion and exclusion criteria now in more detail. The text was re-written as follows:
In this observational longitudinal study, carried out in an academic hospital, we included patients aged 18-65 years who were diagnosed with cutaneous Pso (diagnosed by a dermatologist, with PsA excluded by a rheumatologist PsA (fulfilling Classification Criteria for Psoriatic Arthritis (CASPAR)[12]) and AS (fulfilling Assessment of SpondyloArthritis international Society (ASAS)[13] classification criteria). Patients were selected regardless of presence of clinical ankle enthesitis at the Achilles tendon or plantar fascia. Relevant exclusionary criteria were current use of conventional or synthetic disease-modifying antirheumatic drugs (DMARDs) at time of inclusion and contra-indication for MRI. The current study focuses on longitudinal follow-up with HEMRIS, while baseline MRI-results, in comparison with clinical parameters and PET-CT are described in the previous paper.[14]
- How do you determine the sample size of 38 patients (76 ankles) ? The information about sample size calculation should be described.
Author response: since our study was an exploratory pilot study, no formal sample size calculations were made. We now mentioned this in the ‘Discussion’ section.
- Please offer representative serial images with most apparent changes in "peritendon hypersignal" and "retrocalcaneal bursitis" from your series.
Author response: We thank the reviewer for this excellent suggestion and provided representative images (Figure and 4) in our adjusted manuscript.
- " Changes in clinical disease activity, assessed with measures of general disease activity (Figure 2) and with measures for local disease activity at the enthesis (Figure 3), were 168 not associated with changes in HEMRIS." " Changes in clinical disease activity, in subgroups 169 based on clinical diagnosis of Pso, PsA or AS, were not associated with changes in 170 HEMRIS (Figure 4)". "No difference was observed in HEMRIS results at time of inclusion of 171 the 2 Pso patients who developed PsA during longitudinal follow-up, in comparison to 172 the 9 Pso patients that did not." How do you make sure these findings are not underpowered ?
Author response: we agree with the reviewer that we cannot exclude that a low sample size is the reason we did not find any associations between change in clinical disease activity and change in HEMRIS. We now mention this in the Discussion section. The text was re-written as follows:
One limitation of the study is its relatively small sample size. Since this is an exploratory study, no sample size calculations were made, however this could increase the chance of a type II error. To increase power, we analyzed the data over one-year intervals. With a total follow-up duration of two years, each subject occurred in the dataset twice (regarding comparisons of the HEMRIS with general measures of disease activity), or four times (re-garding comparisons of the HEMRIS with clinical examination of enthesitis at both an-kles). Analyses were not adjusted for multiple measurements in patients because of lim-ited power, however no significant changes in HEMRIS were observed in the unadjusted analyses. Another limitation is the observational study design. Further research in larger populations is required to further establish the responsiveness of HEMRIS to change and its association with change in clinical disease activity. In the ongoing TOFA-PREDICT study (EudraCT Number 2017-003900-28) we aim to investigate the effect of three different DMARDs (tofacitinib, methotrexate and etanercept) on HEMRIS results in a PsA popula-tion with more active disease.
. line 278 "visualizechange" should be corrected.
Author response: we have corrected this error.
Round 2
Reviewer 2 Report
My concerns have been appropriately addressed. It is suitable for publication now.